# Bacterial Isolates from Urinary Tract Infection in Dogs and Cats in Portugal, and Their Antibiotic Susceptibility Pattern: A Retrospective Study of 5 Years (2017–2021)

**DOI:** 10.3390/antibiotics11111520

**Published:** 2022-10-31

**Authors:** Andreia Garcês, Ricardo Lopes, Augusto Silva, Filipe Sampaio, Daniela Duque, Paula Brilhante-Simões

**Affiliations:** 1Inno-Serviços Especializados em Veterinária, R. Cândido de Sousa 15, 4710-300 Braga, Portugal; 2Centre for the Research and Technology of Agro-Environmental and Biological Sciences (CITAB), University of Trás-os-Montes and Alto Douro, Quinta de Prados, 5000-801 Vila Real, Portugal; 3Cooperativa de Ensino Superior Politécnico e Universitário (CRL-CESPU), R. Central Dada Gandra, 1317, 4585-116 Gandra, Portugal

**Keywords:** urinary infection, bacteria, dog, cat, zoonoses, antibiotic resistance

## Abstract

There are growing concerns regarding the rise of antimicrobial-resistant bacteria in companion animals. This study aimed to bring new insights into the current scenario of Portugal’s antimicrobial resistance bacteria isolated from companion animals with urinary tract infections and is the first to be performed during a long period on a large scale. Of a total of 17472 urine samples analyzed, 12,166 (69.6%) (CI 12,200–12,200) were negative for bacterial growth, and 5306 (30.4%) (95% CI 5310–5310) had bacterial growth. Of the culture-positive samples, 5224 (96.6%) (95% CI 5220–5220) were pure cultures and 82 (3.2%) (95% CI 81.9–82.1) had mixed growth. *Escherichia coli* was the most frequently isolated bacteria (*n* = 2360, 44.5%) (95% CI 2360–2360), followed by *Proteus mirabilis* (*n* = 585, 11%) (95% CI 583–583), *Enterococcus faecium* (*n* = 277, 5.2%) (95% CI 277–277) and *Staphylococcus pseudintermedius* (*n* = 226, 4.3%) (95% CI 226–226). The overall susceptibility rates were low for erythromycin (45.3%) and clindamycin (51.3%), and high for aminoglycosides (96.3%), carbapenems (92.4%), trimethoprim-sulfamethoxazole (81.2%), and quinolones (79.9%). *E. coli* also showed considerable resistance to amoxicillin-clavulanic acid. The rates of multidrug-resistant bacteria are still high compared to the northern countries of Europe. This study’s findings show the emergence of antibiotic resistance in the antibiotic agents commonly used in the treatment of UTIs in dogs and cats in Portugal.

## 1. Introduction

Urinary tract infections (UTIs) in companion animals are common and multifactorial [1], generally occurring when host immunity is compromised. The breach in the host defence mechanisms allows an infectious agent (pathogens or opportunistic normal flora) to adhere, multiply, and persist within the urinary tract [2]. A UTI can occur anywhere in the urinary tract, including the bladder (i.e., cystitis) [2]. They can be either endogenous or exogenous. The main cause of UTIs is pathogenic microorganisms, which include bacteria, fungi, protozoa, and viruses [1,2,3]. Age and sex are contributing factors to the incidence of UTIs. In general, bacterial UTIs are more common in dogs than in cats, older animals and females [1,4,5]. Approximately 14% of dogs will develop a bacterial UTI in their lifetime [3,4]. In contrast, UTIs account for approximately 1–3% of all cases of feline lower urinary tract disorders. However, the incidence of bacterial UTIs in cats increases with age [6]. According to the literature, UTIs in companion animals tend to be caused by a single pathogen [1,7]. Studies have shown that the most isolated bacteria from UTIs in dogs were *Escherichia* spp. (45.3%), *Proteus* spp. (13.2%), *Staphylococcus* spp. (11%), and *Enterococcus* spp. (8.6%) [8]. Whereas in the feline population, *Escherichia* spp. (42.7%), *Enterococcus* spp. (22.2%), and *Staphylococcus* spp. (15.2%) were the most frequently isolated bacteria [8].

Antimicrobial resistance (AMR) is a complex problem with many contributing factors. Bacteria have developed numerous methods to resist antibiotic action, such as activation of drug efflux pumps, mutation of antibiotic function sites by passing the target site of the antibiotics, and enzyme-mediated drug degradation [9]. The most important factor is antimicrobial usage (AMU), which facilitates the selection of bacteria with acquired resistance in animals and humans [10,11,12]. Furthermore, resistant bacteria or their resistance genes can be transmitted between animals and humans through direct or indirect contact, food, water, and the environment (e.g., faecal contamination). AMR components can be transferred between bacteria through mobile genetic elements that consist of plasmids, transposons, integrons, and bacteriophages [9]. In the past, the study focus was mainly on food-producing males [7]. With the development of antibiotic resistance in other animals, such as companion animals, it started to limit the treatment options for bacterial infections and has become a concern worldwide. In veterinary practice, the use of empirical antimicrobial treatment of UTI was common, and it has since helped in the development of multidrug-resistant (MDR) organisms, altered normal flora, and colonisation and infection [5]. Other factors are also associated with the development of MDR. MDR is defined by several authors as non-susceptibility to at least one agent in three or more antimicrobial categories and up to (and including) the total number of all antimicrobial categories minus two [13]. Various studies have reported that AMR has increased in companion animal isolates over time and is an emerging problem in public health due to concerns about the zoonotic transmission of AMR [5]. AMR is an important problem in companion animals and should be better understood since it can lead to an increased risk of therapeutic failure, increased treatment costs, and public health problems [14].

This study aimed to bring new insights into the current scenario of Portugal’s antimicrobial pattern in companion animals. In this work, the authors analyze the results from urinary cultures of companion animals (dogs and cats) admitted to the INNO Veterinary Laboratory during the period 2017–2021, to observe: (1) bacterial species isolated; (2) species, breed, sex, and age of the animals affected; and (3) pattern of antibiotic susceptibility and how it has progressed during the period of study.

## 2. Results

### 2.1. Descriptive Data and Bacterial Isolation

Of 17472 urine samples submitted to the INNO Veterinary laboratory between 2017 and 2021, 12,166/17,472 (69.6%) (95% CI 12,200–12,200) were negative for bacterial growth, and 5306/17,472 (30.4%) (95% CI 5310–5310) had bacterial growth. Of the culture-positive samples, 5224/5306 (96.6%) (95% CI 5220–5220) were pure cultures and 82/5306 (3.2%) (95% CI 81.9–82.1) had mixed growth. Figure 1 corresponds to the number of positive and negative urine cultures between 2017 and 2021. In the CFU analysis, 3478/5306 (95% CI 3480–3480) samples were >100 CFU/mL and 827/5306 (95% CI 827–827) samples were <1 CFU/mL. The remaining 1001/5306 (95% CI 1000–1000) samples were >10 UFC/mL. 

Of the 5306 animals with urinary infection, 2730/5306 (51.5%) (95% CI 2730–2730) were canids and 2576/5306 (48,5%) (95% CI 2580–2580) were felines. A total of 3858/5306 (72.7%) (95% CI 3860–3860) animals were males, and the remaining 1448/5306 (27.3%) (95% CI 1450–1450) were females. A total of 91 dog breeds were affected by urinary infection in this study, with dogs without defined breed (SRD) (*n* = 1041), Labrador Retrievers (*n* = 246), and French Bulldogs (*n* = 153) being the most affected. In the case of felines, of the 12 breeds, the most affected were cats without defined breed (SRD) (*n* = 1645), Persian (*n* = 166), and Siamese (*n* = 62). Regarding age, the animals between 6 and 11 years were the most affected (65.2%, *n* = 1702), followed by animals aged 12–20 years (52.9%, *n* = 1385).

A total of 110 bacterial agents were isolated. The main etiologic agent was *Escherichia coli* (44.5%, *n* = 2360/5306) (95% CI 2360–2360), followed by Proteus mirabilis (11%, *n* = 585/5306) (95% CI 585–585). Figure 2 represents the main 10 etiological agents isolated in the positive urine cultures between 2017 and 2021.

Figure 3 represents the trends of *E. coli* during the different years of the study; species, sex, and age of the animals where this agent was isolated. 

### 2.2. Descriptive Antimicrobial Susceptibility Pattern and Multidrug-Resistant Bacteria

A total of 1335/5306 isolates (25%) were classified as MDR. In Table 1, Table 2 and Table 3, the pattern of sensibility in the 5306 in this study is represented. The isolates presented higher resistance percentages to erythromycin (54.7%), ampicillin (50.9%), and penicillin (49.6%) of all the tested antibiotics [15].

### 2.3. Emma’s Categorisation of Antibiotics for Use in Animals

The EMA’s Veterinary Medicines Committee (CVMP) considers the public health risk of antibiotic use in animals and the potential for resistance development. Therefore classification now comprises four categories: category A—”Avoid”, B—“Restrict”, C—“Caution”, and D—“Prudence”. In this paper, we analyse the resistance of antibiotics in these different categories [14].

Figure 4 represents the percentage of resistance to each of the four categories of antibiotics according to Emma’s categorisation [16]. A higher percentage of resistance is observed in the agents from Group D, with 54%, followed by Group B, with 38% resistance. Figure 5 represents the progression of resistance in the four categories of antibiotics between 2017 and 2021. All categories seem to be decreasing, with the exception of category D, where the number of resistances has increased from 2019 to 2021. 

## 3. Discussion

In veterinary practice, inadequate empirical choices, antibiotic treatment of nonbacterial conditions, failure in the administration by the owners, and extensive periods of treatment are the main factors associated with reduced patient outcomes and one of the great contributors to the selection of bacterial resistance [4,11,17]. In this study, the authors analysed the bacterial isolates and their antimicrobial susceptibility patterns from UTIs in dog and cat samples that were admitted to a veterinary laboratory (Inno-Veterinary Laboratory, Braga) during the period from 2017 to 2021. 

A total of 17472 urine samples were analysed, providing us with a large dataset that allowed us to describe the prevalence of bacteria and changes in their antimicrobial resistance over a longer period in companion animals in Portugal. In this study, bacteriuria was detected in 30.4% (5306/17472) of the urinary samples submitted to the laboratory, which was similar to previous observations in other studies in Europe [2,11,18]. Most of the samples had pure growth of a single organism (96.8%), while only 3.2% of the samples contained two or more different organisms. This is also in agreement with previous reports [11,19,20]. The difference between urine samples from canine patients (*n* = 2730/5306, 51.5%) and feline patients (*n* = 2576/5306, 48.5%) is almost inexistent. In other studies, for example, Fonseca et al. [11], the number of urine samples from canine patients was double that of feline patients. These differences can be related to many factors, such as different numbers of pets in different regions or different protocols applied in the clinics. This phenomenon can be explained by the higher prevalence of UTIs in dogs or because the sampling is easily performed in canines [11,20,21]. Concerning the breed, the authors identified no breed predisposition as being the animals with not defined breeds the most affected, both in dogs and cats (dogs = 1041 and cats = 1645) as referred to by other authors [11,22].

The higher rate of UTIs in males (*n* = 3858/5306, 72.7%) than in females (*n* = 1448/5306, 27.3%), corroborates what other authors have described in their studies. In cats, for example, it may be associated with the urethral anatomical conformation of males, which favours the installation of obstructive processes, increasing the risk of infection due to the need to perform the urethral probing procedure [22,23,24]. Regarding age, the animals between 6 and 11 years were the most affected (*n* = 1702/5306, 32.1%), followed by the older animals between 12 and 20 years (26.4%, *n* = 1385). The occurrence of UTI in older animals is expected due to changes in the host defence mechanisms and the appearance of predisposing factors such as chronic renal failure, diabetes mellitus, hyperthyroidism, urinary bladder distention, the presence of uroliths, prolonged use of medications, such as steroidal anti-inflammatory drugs, and urinary incontinence [23,25]. This data is compatible with what has been described in other studies, regardless of sex. Dogs of 7 years of age and cats of 11 years of age, were the animals the most affected by UTIs [11,26,27].

Urinary infections of bacterial origin can be caused by both Gram-positive and Gram-negative bacteria. In the present study, the main etiologic bacteria were *Escherichia coli* (44.5%, *n* = 2360/5306), followed by *Proteus mirabilis* (11%, *n* = 585/5306). In various studies, *E. coli* has been recognised as the most common bacterial cause of UTIs, both in dogs and cats, as observed in our study [11,23,26]. The selection of a suitable antimicrobial for treatment depends in large part on the sensitivity of the organism isolated [27]. In the present study, most isolates were found to be resistant to at least one different group of antibiotics. The isolates presented higher resistance percentages to erythromycin (54.7%), ampicillin (50.9%), and penicillin (49.6%) than all the other tested antibiotics. In veterinary practice, amoxicillin is recommended as a first-line choice for the treatment of UTIs in domestic animals due to its oral bioavailability [28]. Therefore, due to the higher use of this antibiotic, it was expected that many animals would become refractory to amoxicillin and other beta-lactams [11]. Our study observed high levels of resistance to ampicillin (50.9%), penicillin (49.6%), and amoxicillin (42.1%). This is in line with what has been observed in other studies since they are one of the first-line antibiotics used to treat UTIs [11]. The authors expect higher resistance to amoxicillin and not ampicillin, similar to other studies [22,27,28]. This difference can be explained because amoxicillin, in some cases, is not tested in the cards of Vitek 2 Compact in some Gram-negative bacteria. Aminoglycosides were the agents that showed lower resistance percentages. Among aminoglycosides, neomycin (1.2%) and amikacin (1.1%) present a higher susceptibility pattern, with over 90%. This can be attributed to the reduced use of these antibiotics in practice due to their nephrotoxic secondary effects [22,27,28]. Trimethoprim-sulfonamide also presented lower resistance percentages (18.8%). This antibiotic is also a first-line option agent used for the empirical treatment of uncomplicated UTIs in domestic animals and can be a useful alternative to amoxicillin [28]. The results obtained are similar to other studies [11,20,22]. Concerning cephalosporins in general, the percentage of susceptibility is high when compared with other antibiotic groups as described in several works [28,29,30], except for cephalexin, with 48.9% resistance. The high levels of resistance to cephalexin have been reported in other studies since it is one of the antibiotics of choice to treat *Enterococcus* spp. UTIs [28,29,30].

A total of 25% of the isolates (*n* = 1335/5306) were classified as MDR. This percentage is a little high when compared to other countries in Europe with less than 10 %, although Portugal, Spain, and Italy have already been reported as having higher levels of MDR when compared to other European countries [18]. Overall, in the present study, we observed a higher percentage of resistance to Category D according to Emma’s categorisation, and these levels of resistance have been increasing since 2019. This phenomenon was expected since category D—“Prudence” includes antibiotics such as amoxicillin and trimethoprim-sulfamethoxazole that are used as the first line in the treatment of UTI, both in empiricall treatments and in cases were susceptibility tests were performed. However, a decrease in the antibiotics resistance from the categories: A—”Avoid”, B—“Restrict”, and C—“Caution” was observed. This can be associated with many factors, such as the prohibition of the use of agents of category A in veterinary medicine, better training of the clinicians, and the increase of routine susceptibility tests before initiating the treatment [16]. 

This work is a retrospective study and therefore suffers from some limitations. In many cases, the information regarding the method of urine collection and information about whether the animals were neutered or not was due to a lack of clinical information. In many cases, complementary exams such as haematology or biochemistry were not available or were not performed. Antimicrobial susceptibility testing was performed according to the available existing guidelines, so there is the possibility of some differences between the years since its interpretation according to the new guidelines (CLSI VET02, 2021) was not possible retrospectively. Additionally, antibiotic tests that were not present on the Vitek 2 AST-GP71 and AST-GN98 cards (bioMérieux, Marcy l’Etoile, France) were not included. 

## 4. Materials and Methods

Urine specimens from canine and feline patients between January 2017 and December 2021 were submitted for microbiological cultures and respective antibiotic resistance to the Inno Veterinary Laboratory (Braga, Portugal). The samples were collected from diverse veterinary practices in Portugal (continent and isles) by cystocentesis, catheterization, or midstream catch. Information concerning the species, age, sex, and breed was compiled from each animal. The age of the animals was categorised into three groups: 0–5, 6–11, and 12–20 years.

Urine samples were cultured routinely on Chromid CPS Elit agar (bioMérieux, Marcy l’Etoile, France), using a standard inoculation loop that delivers 0.01 mL or 0.001 mL. After incubation at 37 °C for 18–24 h, the plates were examined to determine whether the cultures were pure or mixed based on colony colour and morphology. The approximate number of colony-forming units/millilitre (CFU/mL) of urine was determined for each specimen. The Gram colouration of the colonies was performed to determine the morphology (cocos, bacillus, coccobacillus) of the bacteria and their purity, using the Previcolor System (bioMérieux, Marcy l’Etoile, France). Phenotype identification of the bacteria/pathogens was performed using the automated system Vitek 2 Compact system (bioMérieux, Marcy l’Etoile, France) with the Vitek 2 ID GN and GP cards (refs 21341 and 21342, bioMérieux, Marcy l’Etoile, France). Automated antimicrobial susceptibility testing was performed with the Vitek 2 Compact system method (bioMérieux, Marcy l’Etoile, France), using the Vitek 2 AST-GP71 and AST-GN98 cards (bioMérieux, Marcy l’Etoile, France). The manufacturer’s specifications were followed using minimum inhibitory concentration (MIC) testing, automatically interpreted based on the guidelines provided by the Clinical Laboratory Standards Institute (CLSI) (available in the CLSI VET01-S2 document). The antibiotics tested were imipenem, cefovecin, cefpodoxime, ceftiofur, cephalothin, enrofloxacin, marbofloxacin, amikacin, gentamicin, neomycin, clindamycin, erythromycin, nitrofurantoin, penicillin, amoxicillin, amoxicillin + clavulanic acid, ampicillin, oxacillin, trimethoprim-sulfamethoxazole, tetracyclines, and doxycycline.

For this study, bacteria were also categorised as MDR. The Antimicrobial Advice, Ad Hoc Expert Group (AMEG) and the EMA’s Veterinary Medicines Committee (CVMP) 2019 published a new classification of antibiotics. This new classification considers the public health risk of their use in animals and the potential for resistance development. The classification now comprises four categories: A—”Avoid”, B—“Restrict”, C—“Caution”, and D—“Prudence”. In this paper, we analyse the resistance of antibiotics in these different categories [16]. 

## 5. Conclusions

This work brings new insights into the current scenario of Portugal’s antimicrobial resistance bacteria isolated from companion animals with UTIs and is the first to be performed during a long period on a large scale. All breeds were susceptible to acquiring urinary infections, and older animals were the most predisposed. *E. coli* was the most common agent, and considerable resistance to erythromycin and amoxicillin-clavulanic acid was observed. The rates of MDR are still high compared to the northern countries of Europe. This study’s findings show the emergence of resistance to the antibiotics most commonly used in the treatment of UTIs in dogs and cats in Portugal.

The results of the present study reinforce the importance of performing antimicrobial susceptibility tests, as well as revive the relevance of the veterinarian’s role in the prevention and control of animal UTIs to minimize the spread of bacterial resistance and its impact on the environment, animal, and human health, under the “One Health” concept.

## Figures and Tables

**Figure 1 antibiotics-11-01520-f001:**
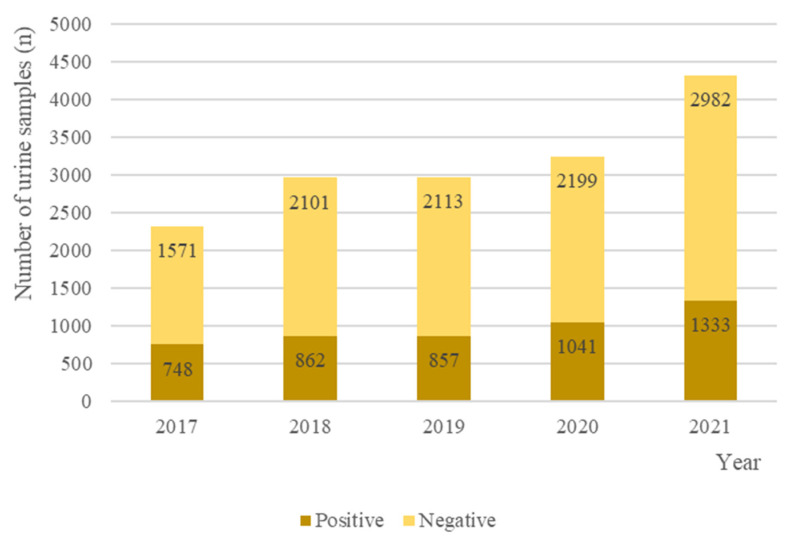
Negative and positive cultures grown from the 17472 urine samples from dogs and cats submitted to the INNO veterinary laboratory between 2017 and 2021.

**Figure 2 antibiotics-11-01520-f002:**
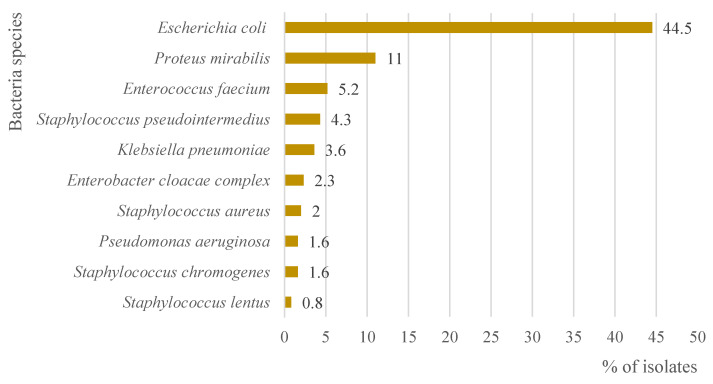
Bacteria species that predominate in the 5306 isolates with positive culture grown from urine samples from dogs and cats submitted to the INNO veterinary laboratory between 2017 and 2021.

**Figure 3 antibiotics-11-01520-f003:**
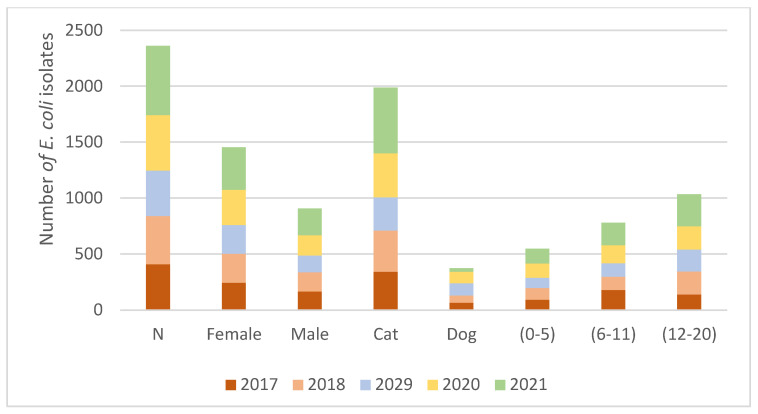
Several *E. coli* isolates were found in every year of the study (2017–2021). The species (dog and cat), sex (female and male), and age (0–5, 6–11, and 12–20 years) in which *E. coli* was isolated by year.

**Figure 4 antibiotics-11-01520-f004:**
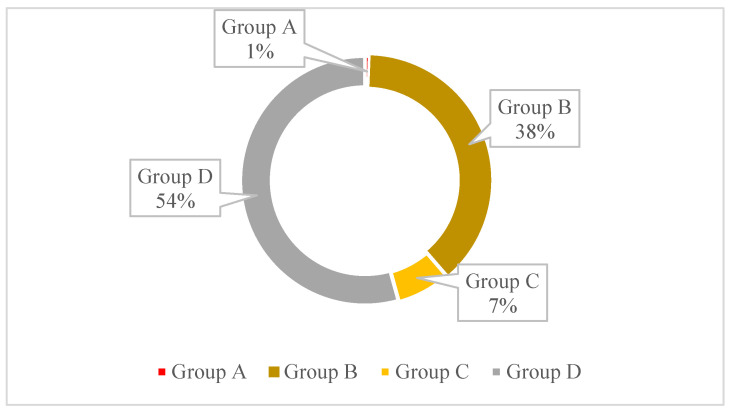
Percentage of resistance in each of the four antibiotic categories according to Emma’s categorisation, in the total of 1335 isolates. The classification comprises four categories: category A—”Avoid”, B—”Restrict”, C—“Caution”, and D—“Prudence”.

**Figure 5 antibiotics-11-01520-f005:**
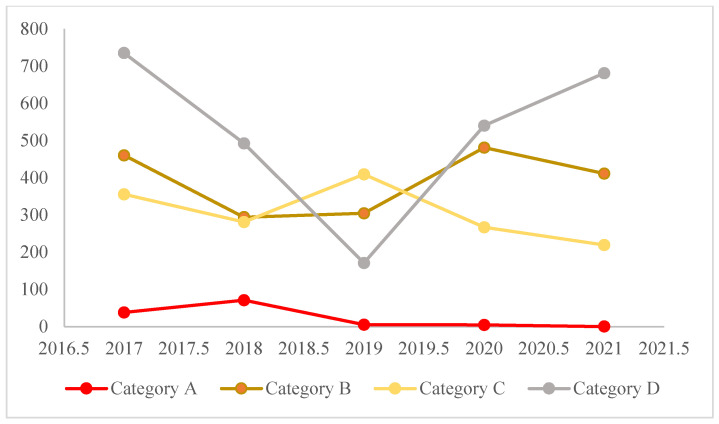
Progression of the resistance in the four antibiotic categories according to Emma’s categorisation of the 1335 isolates between 2017 and 2021.

**Table 1 antibiotics-11-01520-t001:** The pattern of antimicrobial sensibility to carbapenem, cephalosporins, and quinolones in the 5306 isolates between 2017 and 2021 (R—resistant, I—I-intermediate, S—sensible).

	R	I	S	Total	% Resistance
**Carbapenem**					
Imipenem	106	144	1147	1397	7.6
**Cephalosporins**					
Cefovecin	1119	110	3440	4669	24.0
Cefpodoxime	583	23	2928	3534	16.5
Ceftiofur	710	75	3280	4065	17.5
Cephalexin	1649	116	1609	3374	48.9
Cephalothin	865	300	1191	2356	36.5
**Quinolones**					
Enrofloxacin	954	278	3642	4874	19.6
Marbofloxacin	1048	197	3815	5060	20.7

**Table 2 antibiotics-11-01520-t002:** The pattern of antimicrobial sensibility in the 5306 isolates between 2017 and 2021 (R—resistant, I—I-intermediate, S—sensible) to aminoglycosides, glycosides, macrolides, and nitrofurans.

	R	I	S	Total	% Resistance
**Aminoglycosides**					
Amikacin	36	70	3170	3276	1.1
Gentamicin	248	47	3656	3951	6.3
Neomycin	33	77	2091	2201	1.2
**Glycosamides**					
Clindamycin	381	19	383	783	48.7
**Macrolides**					
Erythromycin	573	131	350	1054	54.7
**Nitrofurans**					
Nitrofurantoin	948	250	3430	4628	20.5

**Table 3 antibiotics-11-01520-t003:** The pattern of antimicrobial sensibility in the 5306 isolates between 2017 and 2021 (R—resistant, I—I-intermediate, S—sensible) to nitrofurans, penicillin, sulphonamides, and tetracyclines.

	R	I	S	Total	% Resistance
**Nitrofurans**					
Nitrofurantoin	948	250	3430	4628	20.5
**Penicillin**					
Amoxicillin	671	1	920	1592	42.1
Amoxicillin + clavulanic acid	1027	136	3085	4248	24.2
Ampicillin	1891	82	1735	3708	50.9
Oxacillin	257	0	397	654	39.3
Penicillin	646	2	654	1302	49.6
**Sulphonamides**					
Trimethoprim-sulfamethoxazole	839	20	3602	4461	18.8
**Tetracyclines**					
Doxycycline	1641	233	2761	4635	35.4
Tetracycline	1949	90	2571	4610	42.3

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
