# Peer review of "Bacterial Isolates from Urinary Tract Infection in Dogs and Cats in Portugal, and Their Antibiotic Susceptibility Pattern: A Retrospective Study of 5 Years (2017–2021)"

_antibiotics, 2022, doi:10.3390/antibiotics11111520_

Round 1

Reviewer 1 Report

This study analyzed the bacterial pathogens isolated from the urinary tract of companion animals including dogs and cats for 5 years in Portugal. Overall, the data were well analyzed and presented. However, some revisions are needed.

Groups A-D should be defined in the legend of Figure 3.

Abbreviations, such as UTI in the abstract (define here) and in main text line 54 (removed due to it being defined at the beginning of the introduction).

All the bacterial species should be in italic in the context, such as E. coli.

Grammar checking, lines 23-24, line 65: bacteria specie isolated most common, and line 84: bacterias (not correct, should be bacteria); number expression line 174, 48,5% >48.5%.
Line 182, canines as felines, is not correct.

Gram-negative or gram negative should be consistent in the manuscript.

Finally, antibiotic resistance reasons are not mentioned in the introduction or discussion, which could be another essential factor for treatment, such as carrying antibiotic resistance genes or plasmids (PMID: 35326812).

Author Response

This study analyzed the bacterial pathogens isolated from the urinary tract of companion animals including dogs and cats for 5 years in Portugal. Overall, the data were well analyzed and presented. However, some revisions are needed.

  1. Groups A-D should be defined in the legend of Figure 3.

Author answer: this information was added to the manuscript.

  1. Abbreviations, such as UTI in the abstract (define here) and in main text line 54 (removed due to it being defined at the beginning of the introduction).

Author answer: removed as requested.

  1. All the bacterial species should be in italic in the context, such as E. coli.

Author answers: all the bacteria name were corrected in the text and changed to italic.

  1. Grammar checking, lines 23-24, line 65: bacteria specie isolated most common, and line 84: bacterias (not correct, should be bacteria); number expression line 174, 48,5% >48.5%.

Author answer: the grammar and mistakes were corrected.

  1. Line 182, canines as felines, is not correct.

Author answers: changed to dogs and cats.

  1. Gram-negative or gram negative should be consistent in the manuscript.

Author answers: it was changed to Gram-negative in all the manuscripts.

  1. Finally, antibiotic resistance reasons are not mentioned in the introduction or discussion, which could be another essential factor for treatment, such as carrying antibiotic resistance genes or plasmids (PMID: 35326812).
  2. Author answers: this information was added to the introduction line 46 to 57.

Reviewer 2 Report

A Retrospective Study of 5 Years (2017-2021) showed the emergence of resistance to antibiotics commonly used in the treatment of UTIs in dogs and cats in Portugal. The topic is timely and will be of interest to the readers. However, there are a few minor issues that should be addressed;

1. The title "Bacterial Isolates from Urinary Tract Infection in Dogs and Cats 2 in Portugal, and Their Antibiotic Susceptibility Patter". Please check the word "patter". I think it is Pattern. Correct the title

2. The bacterial names should be italicized throughout the manuscript.

3. Line 24. "Antibiotics most used in the treatment of UTIs in dogs and cats in Portugal". The word "most" should be replaced with "commonly". 

4. Line 64: "INNO" The word INNO should be written in full.

5. Line 65: "1) bacteria specie isolated most common". add "l" with bacteria (bacterial)

6. Line 96: " tetracyclines, doxycycline, and tetracycline" The word "tetracycline is repetition. 

7. Line 117-118: "A total of 9315 (76.6%) animals were males and the remaining 2851 23,4%) were females." Kindly confirm the results mentioned here.

8. Line 125: "A total of 110 different agents were isolated." What do you mean by 110 different agents? Do you mean antibiotics please explain and revise the sentence?

9. Line 125-128: Bacterial names should be italicized and also add the total number of samples. For example (44.5%, n =2360/???) 

10. Table 1: Cephalosporins" I did not find any explanation about Cephalosporins in your discussion section.

11. Line 145: " Emma's categorization" Insert the reference

12. Line 161: Add this reference DOI: 10.1111/jam.15469

13. Will be better to merge some paragraphs in the discussion part. As currently there are several small paras. You should combine some.

Author Response

A Retrospective Study of 5 Years (2017-2021) showed the emergence of resistance to antibiotics commonly used in the treatment of UTIs in dogs and cats in Portugal. The topic is timely and will be of interest to the readers. However, there are a few minor issues that should be addressed;

  1. The title "Bacterial Isolates from Urinary Tract Infection in Dogs and Cats 2 in Portugal, and Their Antibiotic Susceptibility Patter". Please check the word "patter". I think it is Pattern. Correct the title

Author answer: the title was corrected to pattern.

  1. The bacterial names should be italicized throughout the manuscript.

Author answer: the name of the bacteria was corrected to italic in all the manuscript.

  1. Line 24. "Antibiotics mostused in the treatment of UTIs in dogs and cats in Portugal". The word "most" should be replaced with "commonly". 

Author answer: the sentence was changed to “Our findings show the emergence of antibiotic resistance on the antibiotic agents commonly used in the treatment of UTIs in dogs and cats in Portugal.”

  1. Line 64: "INNO" The word INNO should be written in full.

Author answer: The name INNO is not an abbreviation, it is the name of the laboratory in full. 

  1. Line 65: "1) bacteria specie isolated most common". add "l" with bacteria (bacterial)

Author answer: corrected.

  1. Line 96: " tetracyclines, doxycycline, and tetracycline" The word "tetracycline is repetition. 

Author answer: the repeated word was eliminated.

  1. Line 117-118: "A total of 9315 (76.6%) animals were males and the remaining 2851 23,4%) were females." Kindly confirm the results mentioned here.

Author answer: the results were confirmed. It was corrected to “A total of 3858 (72.7%) animals were males and the remaining 1448 (27,3%) were females.”

  1. Line 125: "A total of 110 different agents were isolated." What do you mean by 110 different agents? Do you mean antibiotics please explain and revise the sentence?

Author answer: the authors mean bacterial agents. That information was added to the text.

  1. Line 125-128: Bacterial names should be italicized and also add the total number of samples. For example (44.5%, n =2360/???) 

Author answer: all the names of the bacteria were italicized, and the author apologised for the mistake. In the samples, the total number was added.

  1. Table 1: Cephalosporins" I did not find any explanation about Cephalosporins in your discussion section.

Author answer: This information was added to the discussion section.

  1. Line 145: " Emma's categorization" Insert the reference

Author answer: reference added.

  1. Line 161: Add this reference DOI: 10.1111/jam.15469

Author answer: reference added.

  1. Will be better to merge some paragraphs in the discussion part. As currently there are several small paras. You should combine some

Author answer: some of the paragraphs were merged in the discussion has requested.

Reviewer 3 Report

The authors in this manuscript have scientific merit that warrants publication in Antibiotics (A Retro-3 spective Study of 5 Years of Bacterial UTIs in companion animals in Portugal). The authors have pointed out firstly the prevalence of Bacterial UTIs and in addition the alarming emergence of multi drug resistance in the treatment of UTIs in dogs and cats in Portugal. Overall, the study undertaken by the authors is relevant and significant to the importance of the role of the veterinarian in the prevention and control of animal UTIs, and depending on a new classification of antibiotics by EMA (2019). These will help the veterinarian in optimizing guidelines for using antibiotics. However, the manuscript needs to be rewritten for some sentences, and the scientific names of bacterial genera and species should be printed in italics as well as the addition of a legend in a few Figures are needed. In addition, it would be better to have a native English speaker review the English language of this manuscript extensively.

Minor comments:

Abstract:

Line 15, 16 & 17 need to be rewritten because you mentioned that 5306 of bacterial growth but it was 4989 isolated bacteria.

                                       “and 5306 (30.4%) had bacterial growth. From the positive samples, a total of 4989 agents were isolated. Of the culture-positive samples, 5224 (96.6%) were pure cultures and 82 (3.2%) had mixed growth.”

Line 23 needs more clarification about the conclusion

“Our findings show the emergence of resistance to 23 antibiotics most used in the treatment of UTIs in dogs and cats in Portugal”

Line 16: bacteria,   instead of agents

Line 21: E. coli,    instead of E. coli

Line 42: the most isolated bacteria was, instead of the agent most isolated

Lines 42-45: Escherichia spp,    instead of Escherichia spp

                    Proteus spp,   instead of Proteus spp

      Staphylococcus spp. ,    instead of Staphylococcus spp.

     Enterococcus spp.,    instead of Enterococcus spp.

Line 52 rewrite For a long time, the focus was mainly on animals used for food production [7].

Line 53: UTI, instead of urinary tract infections

Line 74: sex,     remove coma “,”

Line 84: remove "s"  bacteria,    instead of bacterias

Line 85: Phenotype identification of the bacteria / pathogens,  instead of Phenotype identification of the agents

Line 87-95: The sentence is too lengthy. Concise it

Line 99-102: The sentence is too lengthy. Concise it

Line 110-111: it is not clear Sentence and Please revise the sentence

Line 110-111:

Can you explain more why “a total of 4989 agents were isolated”? Is there any difference from “5224 (96.6%) were pure cultures and  (3.2%) had mixed growth”

Line 116-123: try to present data in extra Table s

Line 125: The main etiologic bacteria was, instead of The main etiologic agent

Lines 125-126: E.  coli,    instead of Escherichia coli

                    P. mirabilis,   instead of Proteus mirabilis

Lines 125-126: E.  coli,    instead of Escherichia coli

                    P. mirabilis,   instead of Proteus mirabilis

Line 130: Revise the title of Figure 1 and add a legend as well.

Line 133: Revise the title of Figure 2 and add the legend as well. Perhaps the authors can improve the figure by presenting data as a % with the number.

Line 135: 3.2. Antimicrobial susceptibility pattern  Perhaps the authors can try to add explain more

Line 143: 3.3. Emma’s categorization of antibiotics for use in animal . Perhaps the authors can try to give an explanation or give the mean of Emma’s categorization in the introduction of MS as well.

Line 153: in Figure 3 it would be better if add a legend as well and explain more details about these 4 categorization groups.

Line 162: UTIs, instead of urinary tract infections

Line 183: UTI, instead of urinary tract infections

Line 192: diabetes mellitus, instead of Diabetes Mellitus

Line 197: by UTIs, instead of by urinary infections

Line 199: The main etiologic bacteria was,   instead of the main etiologic agent was

Line 200: E. coli,    instead of E. coli

Line 214: What are studies you mean here and add the references similar to other studies” Add

Line 177, 226, and 233: “But” , change to formal academic writing or revise the sentences

Line 227: MRS ,  is you mean MDR?, change it.

Line 251: E. coli was the most  (italic) instead of E. coli was the most common agent,

 Discussion

The authors need to double-check the finding such as  (line 168-169) with other previous reported such as reference 10 reported that less than their finding   “Significant bacteriuria was detected in 18.4% of samples from dogs and 10.0% from cats,”  while your finding was double than them

 Line 168-169 “We detected bacteriuria in 30.4% (5306/17472) of the urinary samples submitted to the laboratory, which was similar to previous observations in other studies in Europe [3,10,14]”

Line 211-212  “penicillin (49.6%)” but they

10. Fonseca, J.D.; Mavrides, D.E.; Graham, P.A.; McHugh, T.D. Results of Urinary Bacterial Cultures and Antibiotic Suscepti-bility Testing of Dogs and Cats in the UK. J. Small Anim. Pract. 2021, 62, 1085–1091, doi:10.1111/jsap.13406.

Line 177-178:  Revise the

“But it is possible to observe in 177 our study the presence of a slightly higher number of samples from the canine source” 

Line 217-220: Please revise the sentence

 References

Although the three references are in Portuguese and the abstract in English, the Authors should follow the guidelines of Antibiotics and need to check and re-write all references in English

 Line 314-315

19. Da, D.R.; Gonçalves, Y.; Ramalho, J.; Lopes, M.A.; Domingues, B.L. ESTUDO RETROSPECTIVO DA ETIOLOGIA, 314 SENSIBILIDADE ANTIBIÓTICA, AVALIAÇÃO HEMATOLÓGICA E BIOQUÍMICA DE INFECÇÕES DO TRATO 315 URINÁRIO DE CÃES E GATOS. 2018, 33, 14

Line 317-320

20. Ferreira, M.C.; Nobre, D.; Oliveira, M.G.X. de; Oliveira, M.C.V. de; Cunha, M.P.V.; Menão, M.C.; Leite Dellova, D.C.A.; 317 Knöbl, T. Agentes bacterianos isolados de cães e gatos com infecção urinária: perfil de sensibilidade aos antimicrobianos. 318 Atas Saúde Ambient. - ASA 2014, 2, 29–37.

21. Reche Junior, A. A orbifloxacina no tratamento das cistites bacterianas em gatos domésticos. Ciênc. Rural 2005, 35, 1325–320 1330, doi:10.1590/S0103-84782005000600015.

Date of this review

11 October 2022

Author Response

The authors in this manuscript have scientific merit that warrants publication in Antibiotics (A Retro-3 spective Study of 5 Years of Bacterial UTIs in companion animals in Portugal). The authors have pointed out firstly the prevalence of Bacterial UTIs and in addition the alarming emergence of multi drug resistance in the treatment of UTIs in dogs and cats in Portugal. Overall, the study undertaken by the authors is relevant and significant to the importance of the role of the veterinarian in the prevention and control of animal UTIs, and depending on a new classification of antibiotics by EMA (2019). These will help the veterinarian in optimizing guidelines for using antibiotics. However, the manuscript needs to be rewritten for some sentences, and the scientific names of bacterial genera and species should be printed in italics as well as the addition of a legend in a few Figures are needed. In addition, it would be better to have a native English speaker review the English language of this manuscript extensively.

Minor comments:

Abstract:

  1. Line 15, 16 & 17 need to be rewritten because you mentioned that 5306 of bacterial growth but it was 4989 isolated bacteria.

                                       “and 5306 (30.4%) had bacterial growth. From the positive samples, a total of 4989 agents were isolated. Of the culture-positive samples, 5224 (96.6%) were pure cultures and 82 (3.2%) had mixed growth.”

Author answer: the sentence was corrected.

  1. Line 23 needs more clarification about the conclusion

“Our findings show the emergence of resistance to 23 antibiotics most used in the treatment of UTIs in dogs and cats in Portugal”

Author answer: The sentence was corrected. “Our findings show the emergence of antibiotic resistance on the antibiotic agents commonly used in the treatment of UTIs in dogs and cats in Portugal.”

  1. Line 16: bacteria,   instead of agents

Author answer: The sentence was corrected.

  1. Line 21:  coli,    instead of E. coli

Author answer: corrected.

  1. Line 42: the most isolated bacteria was, instead of the agent most isolated

Author answer: The sentence was corrected.

  1. Lines 42-45: Escherichia spp,    instead of Escherichia spp

                    Proteus spp,   instead of Proteus spp

      Staphylococcus spp. ,    instead of Staphylococcus spp.

     Enterococcus spp.,    instead of Enterococcus spp.

Author answer: corrected.

  1. Line 52 rewrite For a long time, the focus was mainly on animals used for food production [7].

Author answer: The sentence was corrected.

  1. Line 53: UTI, instead of urinary tract infections

Author answer: corrected.

  1. Line 74: sex,     remove coma “,”

Author answer: corrected.

  1. Line 84: remove "s"  bacteria,    instead of bacterias

Author answer: corrected.

  1. Line 85: Phenotype identification of the bacteria / pathogens,  instead of Phenotype identification of the agents

Author answer: corrected.

  1. Line 87-95: The sentence is too lengthy. Concise it

Author answer: corrected.

  1. Line 99-102: The sentence is too lengthy. Concise it

Author answer: corrected.

  1. Line 110-111: it is not clear Sentence and Please revise the sentence

Author answer: corrected.

  1. Line 110-111:

Can you explain more why “a total of 4989 agents were isolated”? Is there any difference from “5224 (96.6%) were pure cultures and  (3.2%) had mixed growth”

Author answer: The sentence was deleted.

  1. Line 116-123: try to present data in extra Table s

Author answer: The data was presented in extra tables as requested.

  1. Line 125: The main etiologic bacteria was, instead of The main etiologic agent

Author answer: corrected.

  1. Lines 125-126: coli,    instead of Escherichia coli
  2. mirabilis,   instead of Proteus mirabilis

Author answer: corrected.

  1. Lines 125-126: coli,    instead of Escherichia coli
  2. mirabilis,   instead of Proteus mirabilis

Author answer: corrected.

  1. Line 130: Revise the title of Figure 1 and add a legend as well.

Author answer: the title and legend were corrected.

  1. Line 133: Revise the title of Figure 2 and add the legend as well. Perhaps the authors can improve the figure by presenting data as a % with the number.

Author answer: the title and legend were corrected and the % was added.

  1. Line 135: 2. Antimicrobial susceptibility pattern  Perhaps the authors can try to add explain more

Author answer: the title was improved

  1. Line 143: 3. Emma’s categorization of antibiotics for use in animal . Perhaps the authors can try to give an explanation or give the mean of Emma’s categorization in the introduction of MS as well.

Author answer: this information was added in this section.

  1. Line 153: in Figure 3 it would be better if add a legend as well and explain more details about these 4 categorization groups.

Author answer: information added.

  1. Line 162: UTIs, instead of urinary tract infections

Author answer: corrected.

  1. Line 183: UTI, instead of urinary tract infections

Author answer: corrected.

  1. Line 192: diabetes mellitus, instead of Diabetes Mellitus

Author answer: corrected.

  1. Line 197: by UTIs, instead of by urinary infections

Author answer: corrected.

  1. Line 199: The main etiologic bacteria was,   instead of the main etiologic agent was

Author answer: corrected.

  1. Line 200:  coli,    instead of E. coli

Author answer: corrected.

  1. Line 214: What are studies you mean here and add the references “similar to other studies” Add

Author answer: added.

  1. Line 177, 226, and 233: “But” , change to formal academic writing or revise the sentences

Author answer: added.

  1. Line 227: MRS ,  is you mean MDR?, change it.

Author answer: added.

  1. Line 251:  coliwas the most  (italic) instead of E. coli was the most common agent,

Author answer: corrected.

  1. Discussion

The authors need to double-check the finding such as  (line 168-169) with other previous reported such as reference 10 reported that less than their finding   “Significant bacteriuria was detected in 18.4% of samples from dogs and 10.0% from cats,”  while your finding was double than them

 Line 168-169 “We detected bacteriuria in 30.4% (5306/17472) of the urinary samples submitted to the laboratory, which was similar to previous observations in other studies in Europe [3,10,14]”

Line 211-212  “penicillin (49.6%)” but they

  1. Fonseca, J.D.; Mavrides, D.E.; Graham, P.A.; McHugh, T.D. Results of Urinary Bacterial Cultures and Antibiotic Suscepti-bility Testing of Dogs and Cats in the UK. J. Small Anim. Pract. 2021, 62, 1085–1091, doi:10.1111/jsap.13406.

 Author answer: this section of the text was confirmed and corrected.

  1. Line 177-178:  Revise the

“But it is possible to observe in 177 our study the presence of a slightly higher number of samples from the canine source” 

Author answer: sentence deleted.

Line 217-220: Please revise the sentence

 References

Although the three references are in Portuguese and the abstract in English, the Authors should follow the guidelines of Antibiotics and need to check and re-write all references in English

 Line 314-315

  1. Da, D.R.; Gonçalves, Y.; Ramalho, J.; Lopes, M.A.; Domingues, B.L. ESTUDO RETROSPECTIVO DA ETIOLOGIA, 314 SENSIBILIDADE ANTIBIÓTICA, AVALIAÇÃO HEMATOLÓGICA E BIOQUÍMICA DE INFECÇÕES DO TRATO 315 URINÁRIO DE CÃES E GATOS. 2018, 33, 14

Line 317-320

  1. Ferreira, M.C.; Nobre, D.; Oliveira, M.G.X. de; Oliveira, M.C.V. de; Cunha, M.P.V.; Menão, M.C.; Leite Dellova, D.C.A.; 317 Knöbl, T. Agentes bacterianos isolados de cães e gatos com infecção urinária: perfil de sensibilidade aos antimicrobianos. 318 Atas Saúde Ambient. - ASA 2014, 2, 29–37.
  2. Reche Junior, A. A orbifloxacina no tratamento das cistites bacterianas em gatos domésticos. Ciênc. Rural 2005, 35, 1325–320 1330, doi:10.1590/S0103-84782005000600015.

Author answer: all the references were corrected according to the guidelines of journal.

Reviewer 4 Report

The manuscript “Bacterial Isolates from Urinary Tract Infection in Dogs and Cats in Portugal, and Their Antibiotic Susceptibility Patter: A Retrospective Study of 5 Years (2017-2021)” was revised. The paper is interesting for Antibiotics and for the special issue “Drugs for Superbugs: Antibiotic Discovery, Modes of Action and Mechanisms of Resistance”. However, authors can improve the quality of the manuscript in many points. Also, a deep discussion should be conducted, not only focused on comparing the data. According to the guidelines of MDPI (https://www.mdpi.com/about/article_types), this manuscript should be classified as a Brief Report. 

Comments:

1) Adjust the italics of the scientific names (lines 21, 43, 44, 45, 125, 126, 200, and many others)

2) Avoid use abbreviated keywords (for example, UTI) and repeated keywords from the title (dog, cat)

3) Line 53-54: UTI was previously defined in line 29.

4) Line 72-73: “The samples were collected from diverse veterinary practices in Portugal (continent and isles)”. Should  be interesting include a map with the location of each sample (group) and the number of sample collected in each region/state/city.

5) How animals were selected?

6) Why the timespan between 2017-2021 was selected? Should be interesting to compare the data each year?

7) The data (figures 1 and 2) should be discriminated between dog, cat, male, female, age.

8) Line 117-118: The decimals should be revised. Example: 48,5 should be 48.5

9) Avoid use “we/our”. The article should be neutral.

10) Line 175. Adjust the reference. Fonseca et al., 2021 should have a number instead of the year.

11) Line 183-184: “Regarding the sex of the animals studied, males had a higher rate of urinary tract infection (n=9315, 60%) than females (n=2851, 23.4%)”. Is this a results obtained or comparing? This data was presented in the Discussion, and should be moved to Results. I suggest using a section with “Results and Discussion”.

12 ) English should be revised and improved.

Author Response

The manuscript “Bacterial Isolates from Urinary Tract Infection in Dogs and Cats in Portugal, and Their Antibiotic Susceptibility Patter: A Retrospective Study of 5 Years (2017-2021)” was revised. The paper is interesting for Antibiotics and for the special issue “Drugs for Superbugs: Antibiotic Discovery, Modes of Action and Mechanisms of Resistance”. However, authors can improve the quality of the manuscript in many points. Also, a deep discussion should be conducted, not only focused on comparing the data. According to the guidelines of MDPI (https://www.mdpi.com/about/article_types), this manuscript should be classified as a Brief Report. 

Comments:

  • Adjust the italics of the scientific names (lines 21, 43, 44, 45, 125, 126, 200, and many others)

Author answer: corrected.

  • Avoid use abbreviated keywords (for example, UTI) and repeated keywords from the title (dog, cat)

Author answer: corrected

  • Line 53-54: UTI was previously defined in line 29.

Author answer: corrected.

  • Line 72-73: “The samples were collected from diverse veterinary practices in Portugal (continent and isles)”. Should  be interesting include a map with the location of each sample (group) and the number of sample collected in each region/state/city.

Author answer: although interesting was not possible to perform this map because the information is not always available. Sometimes is the veterinarians themselves and not the clinic that send them, and the region is not available in the data.

5) How animals were selected?

Author answer: all samples of urine that were admitted to the Service of Microbiology for routine microbiology were selected. From those samples only samples originating from dogs and cats were included, other animals such as horses and exotic pets were eliminated.

6) Why the timespan between 2017-2021 was selected? Should be interesting to compare the data each year?

Author answer: only these years were included because was after 2017 that the data from the routine samples were digitalized and the software “Clinidata” was used to manage all the samples. Before 2017 the data is available but is missing some information such as breed, sex, and some antibiotics. Therefore the authors choose to use only 2017 to 2021, which were the years with more complete data. Also, the laboratory started in 2014, and till 2015-2016 did not perform bacteriology. 

7) The data (figures 1 and 2) should be discriminated between dog, cat, male, female, age.

Author answer: due to the complexity of the presentation of the data in the form of graphics, the authors should present the data requested only for the E. coli isolates, that correspond to the majority of the samples analysed.

8) Line 117-118: The decimals should be revised. Example: 48,5 should be 48.5

Author answer: corrected.

9) Avoid use “we/our”. The article should be neutral.

Author answer: corrected.

10) Line 175. Adjust the reference. Fonseca et al., 2021 should have a number instead of the year.

Author answer: corrected.

11) Line 183-184: “Regarding the sex of the animals studied, males had a higher rate of urinary tract infection (n=9315, 60%) than females (n=2851, 23.4%)”. Is this a results obtained or comparing? This data was presented in the Discussion, and should be moved to Results. I suggest using a section with “Results and Discussion”.

Author answer: the sentence was correct. 

12 ) English should be revised and improved

Author answer: as suggested the english was improved.

Reviewer 5 Report

The manuscript by Garcês et al. present interesting data in the understanding of antimicrobial resistance pattern of urinary tract origin bacterial pathogens from pets. However, before its further processing there are several concerns that need to be addressed, as outlined below:

Line 13: „UTIs” please avoid the using directly acronyms

Line 14: replace „submitted” with „analyzed”

Line 15: „69.6%” – when you express overall prevalence values, pleas insert the value of computed 95% Confidence Interval

Line 20: „96.25%” – please express prevalence values with a single decimal

Line 21: „E. coli” – please revise the italics writing of the scientific name of all species throughout the manuscript– MAJORN CONCERN

Line 29: „UTIs” instead of „UTI”

Lines 43-44: „Escherichia spp. (45.3%), Proteus spp. (13.2%), Staphylococcus spp. (11%), and Enterococcus spp. (8.6%) [8]” – revise the italics writing of species throughout the manuscript (!)

Line 52: „on food producing animals” instead of „on animals used for food production”

Lines 53-54: „UTIs” instead of „urinary tract 53 infections (UTI)”

Line 55: please define de MDR with appropriate new reference(s)

Line 63: „we analyse” – please avoid the personal mode verb formulations. Please revise this issue throughout the manuscript (!)

Line 63: „companion” instead of „company”

Lines 64-67: please try to rephrase these sentences improving the scientific style

Line 68: according to my knowledge for antibiotics journal the results section must immediately follow the introduction. Please revise this and operate the subsequent changes if it is true!!!

Line 76: the study involves animals please provide any writing consent of the owners

Line 78: when you mention used reagents/equipment’s please uniformly indicate the company name, city and country. So, it will become „(bioMérieux, Marcy l’Etoile, France)”

Line 92: being an European study, I wonder why the authors used CLSI guideline for results interpretation instead of EUCAST? -

Lines 98-99: the MDR definition is incorrect, please revise

Line 104: I wonder, were the results statistically interpreted? Being an epidemiological study, this is mandatory! Please complete the study with data presenting the statistical analysis results between the enrolled epidemiological factors (species, age, sex, and breed) – MAJOR CONCERN!

Lines 125-126: italics of scientific names of the species

Line 136: „In the Table 1...”

Line 136: revise the MDR – THE BIGGEST CONCERN!!!

Line 140: „susceptibility” instead of „sensibility” – revise this throughout the manuscript

Line 141: „intermediate” instead of „intermedia”

Line 198: Gram – with sentence case

Line 268: „Informed Consent Statement: Not applicable” – not true!

Line 271: please carefully revise the inconsistency of the reference list, there are several nonconformities according to the journal requirement (e.g. bold writing of the study year, italics writing of the journal abbreviation, volume and scientific name of the species)

There are several inappropriate formulations and typing mistakes – MAJOR CONCERN!!!

Author Response

The manuscript by Garcês et al. present interesting data in the understanding of antimicrobial resistance pattern of urinary tract origin bacterial pathogens from pets. However, before its further processing there are several concerns that need to be addressed, as outlined below:

  1. Line 13: „UTIs” please avoid the using directly acronyms

Author answer: changed.

  1. Line 14: replace „submitted” with „analyzed”

Author answer: as suggested the english was improved.

  1. Line 15: „69.6%” – when you express overall prevalence values, pleas insert the value of computed 95% Confidence Interval

Author answer: information added

Author answer:

  1. Line 20: „96.25%” – please express prevalence values with a single decimal

Author answer: corrected

  1. Line 21: „ coli” – please revise the italics writing of the scientific name of all species throughout the manuscript– MAJORN CONCERN

Author answer: corrected in all the manuscripts.

  1. Line 29: „UTIs” instead of „UTI”

Author answer: corrected

  1. Lines 43-44: „Escherichia spp. (45.3%), Proteus spp. (13.2%), Staphylococcus spp. (11%), and Enterococcus spp. (8.6%) [8]” – revise the italics writing of species throughout the manuscript (!)

Author answer: corrected

  1. Line 52: „on food producing animals” instead of „on animals used for food production”

Author answer: corrected

  1. Lines 53-54: „UTIs” instead of „urinary tract 53 infections (UTI)”

Author answer: corrected

  1. Line 55: please define de MDR with appropriate new reference(s)

Author answer: MDR was defined with new references..

Author answer: corrected

  1. Line 63: „we analyse” – please avoid the personal mode verb formulations. Please revise this issue throughout the manuscript (!)

Author answer: the issue was corrected all over the manuscript.

  1. Line 63: „companion” instead of „company”

Author answer: corrected

  1. Lines 64-67: please try to rephrase these sentences improving the scientific style

Author answer: corrected

  1. Line 68: according to my knowledge for antibiotics journal the results section must immediately follow the introduction. Please revise this and operate the subsequent changes if it is true!!!

Author answer: the order was corrected

  1. Line 76: the study involves animals please provide any writing consent of the owners

Author answer: the study does not involve the animal owners, is only a study regarding the sample that reach the laboratory.  

  1. Line 78: when you mention used reagents/equipment’s please uniformly indicate the company name, city and country. So, it will become „(bioMérieux, Marcy l’Etoile, France)”

Author answer: corrected

  1. Line 92: being an European study, I wonder why the authors used CLSI guideline for results interpretation instead of EUCAST? –

Author answer: because the VITEK2Compact was defined according to CLSI guidelines when the laboratory acquire it.

  1. Lines 98-99: the MDR definition is incorrect, please revise

Author answer: corrected.

  1. Line 104: I wonder, were the results statistically interpreted? Being an epidemiological study, this is mandatory! Please complete the study with data presenting the statistical analysis results between the enrolled epidemiological factors (species, age, sex, and breed) – MAJOR CONCERN!

Author answer: it is a descriptive study from the sample that reach the laboratory, although in a great number and almost from ecery distrit is not representative of the population in Portugal. As the review suggested the authors try to improve the statistic from the manuscript.

  1. Lines 125-126: italics of scientific names of the species

Author answer: corrected

  1. Line 136: „In the Table 1...”

Author answer: corrected

  1. Line 136: revise the MDR – THE BIGGEST CONCERN!!!

Author answer: corrected

  1. Line 140: „susceptibility” instead of „sensibility” – revise this throughout the manuscript

Author answer: corrected

  1. Line 141: „intermediate” instead of „intermedia”

Author answer: corrected

  1. Line 198: Gram – with sentence case

Author answer: corrected

  1. Line 268: „Informed Consent Statement: Not applicable” – not true!

Author answer: the owners were not involved.

  1. Line 271: please carefully revise the inconsistency of the reference list, there are several nonconformities according to the journal requirement (e. g. bold writing of the study year, italics writing of the journal abbreviation, volume and scientific name of the species)

Author answer: corrected.

  1. There are several inappropriate formulations and typing mistakes – MAJOR CONCERN!!!

Author answer: corrected.

.

Round 2

Reviewer 4 Report

The paper can be accepted.

Reviewer 5 Report

The manuscript present interesting data in the understanding of antimicrobial resistance pattern of urinary tract origin bacterial pathogens from pets.